# Urine and Free Immunoglobulin Light Chains as Analytes for Serodiagnosis of Hantavirus Infection

**DOI:** 10.3390/v11090809

**Published:** 2019-09-01

**Authors:** Satu Hepojoki, Lauri Kareinen, Tomas Strandin, Antti Vaheri, Harry Holthöfer, Jukka Mustonen, Satu Mäkelä, Klaus Hedman, Olli Vapalahti, Jussi Hepojoki

**Affiliations:** 1Department of Virology, Medicum, University of Helsinki, 00290 Helsinki, Finland (S.H.) (L.K.) (T.S.) (A.V.) (K.H.) (O.V.); 2Zentrum für Innere Medizin, Universitätsklinikum Hamburg-Eppendorf, 22547 Hamburg, Germany; 3Department of Internal Medicine, Tampere University Hospital, 33520 Tampere, Finland (J.M.) (S.M.); 4Faculty of Medicine and Health Technology, Tampere University, 33014 Tampere, Finland; 5Virology, Helsinki University Hospital, 00290 Helsinki, Finland; 6Veterinary Biosciences, Veterinary Faculty, University of Helsinki, 00790 Helsinki, Finland; 7Institute of Veterinary Pathology, Vetsuisse Faculty, University of Zürich, 8057 Zürich, Switzerland

**Keywords:** rapid diagnosis, free light chain, hantavirus, serodiagnosis

## Abstract

Rapid point-of-care testing is a megatrend in infectious disease diagnosis. We have introduced a homogeneous immunoassay concept, which is based on the simultaneous binding of antigen and protein L to a given immunoglobulin molecule. The complex formation is detected utilizing time-resolved Förster resonance energy transfer between antigen-attached donor and acceptor-labeled protein L, hence the name LFRET. Here, we demonstrate that urine can be used as a sample matrix in LFRET-based serodiagnostics. We studied urine samples collected during the hospitalization and recovery of patients with acute Puumala orthohantavirus (PUUV) infection. We compared PUUV antibody-specific LFRET signals in urine to those in plasma, and found excellent correlation in the test outcomes The LFRET test from urine was positive in 40/40 patients with acute PUUV infection. PUUV causes a mild form of hemorrhagic fever with renal syndrome, characterized by acute kidney injury and proteinuria. Immunofluorescence and western blotting demonstrated PUUV-IgG and -IgA in urine, however, the presence of intact immunoglobulins did not fully explain the LFRET signals. We purified free light chains (FLCs) from both urine and serum of healthy volunteers and patients with acute PUUV infection, and verified the presence of antigen-specific FLCs. Antigen-specific FLCs provide a new means for non-invasive antibody detection and disease diagnosis.

## 1. Introduction

There is a constant strive towards the development of diagnostic tools for infectious diseases, to reduce the time gap between the symptoms onset and diagnosis. This is not only important for administering appropriate treatment, but also for the avoidance of unnecessary medication. Although rapid, the existing point-of-care (POC) methods tend to lack sensitivity, specificity, or both. We have approached this challenge by applying time-resolved Förster resonance energy transfer (TR-FRET) in the generation of homogeneous immunoassays with diagnostic performances comparable to those of central laboratory reference tests [1,2,3]. An assay referred to as “LFRET” utilizes TR-FRET between a fluorophore-labeled protein L and a fluorophore-labeled microbial antigen bound by a given immunoglobulin (Ig) molecule [4]. The serodiagnosis of acute orthohantavirus infection using serum as a sample material provided proof-of-concept for this approach [3]. Protein L binds the immunoglobulin (Ig) light chain kappa without interfering with its antigen recognition [5]. The kappa to lambda ratio varies among individuals, with an average of 2:1 for intact immunoglobulins. Since the kappa chains are present in all Ig classes [6], the LFRET approach in principle is also applicable to the diagnosis of autoimmune diseases and allergies.

Orthohantaviruses, earlier known as hantaviruses at the genus level, are classified into the order *Bunyavirales*, family *Hantaviridae*, subfamily *Mammantavirinae*, genus *Orthohantavirus*. Many orthohantaviruses are zoonotic and may cause life-threatening infections in humans [7]. While those endemic in the Americas cause hantavirus cardiopulmonary syndrome (HCPS), those in Eurasia cause hemorrhagic fever with renal syndrome (HFRS) of varying severity [7]. Orthohantaviruses establish in their reservoir host a persistent infection, in which the virus is secreted in urine, feces, and saliva [7]. Puumala virus (PUUV) carried by bank voles (*Myodes glareolus*) is an orthohantavirus. endemic in Eurasia with thousands of confirmed annual cases, with the highest per capita incidence rate in Finland [8]. Transmission to man occurs via inhalation of aerosolized rodent excreta, and often manifests as a mild HFRS termed nephropathia epidemica (NE) [9]. While the mortality rate of HFRS caused by Hantaan virus (HTNV) can reach 10%, the mortality rate of PUUV is 0.1–0.5% [7]. The symptoms of PUUV infection include abrupt fever, headache, nausea, and abdominal and back pain. Renal involvement includes proteinuria, hematuria, and acute kidney injury (AKI) [10,11,12]. Transient proteinuria is present in most patients [11]. It begins abruptly, can be major, and consists mainly of albumin. The proteinuria is nonselective, due to the glomerular barrier defect. The concomitant urinary loss of low-molecular-weight proteins such as β2-microglobulin and α1-microglobulin indicates tubular injury [11]. Glomerular (albumin), but not tubular proteinuria, predicts the severity of AKI [13].

Although AKI is considered the hallmark of HFRS, pulmonary, cardiac, and central nervous system symptoms including somnolence, and especially visual disturbances, are common [9,14,15,16]. Serodiagnosis of acute PUUV infection is based on detection of IgM or low-avidity IgG, both of which target the nucleocapsid (N) protein [17,18]. Additional forms of PUUV serodiagnostics include enzyme immunoassay (EIA), immunoblotting, and immunofluorescence assay (IFA). Moreover, an immunochromatographic rapid test for IgM is available [17,18,19]. As the PUUV N protein cross-reacts particularly with HCPS sera, at least two antigens (e.g., PUUV or Sin Nombre and either Dobrava, Seoul, or HTNV nucleocapsid proteins) should be used to cover the spectrum of hantaviruses causing human infection.

While setting up the LFRET assay for PUUV infection, we observed that some of the acute-phase sera elicited strikingly high TR-FRET signals. We hypothesized that in addition to intact immunoglobulins (Igs), free light chains (FLCs) targeting the antigen (PUUV N) might be involved. During Ig biosynthesis, FLCs are produced in excess [20,21,22,23,24] and excreted via urine [25]. It has been shown that FLCs can specifically bind antigens albeit with usually lower affinity than intact Igs [21]; to our knowledge this has not been utilized in diagnosis. Namely, the potential of LFRET to detect antimicrobial FLCs prompted us to examine the utility of urine as a sample matrix in infectious disease serodiagnostics.

## 2. Materials and Methods

### 2.1. Clinical Samples and Ethics Statement

We studied a panel of archival (−80 °C) plasma and urine samples collected from 40 patients hospitalized due to acute PUUV infection at Tampere University Hospital during 2005 to 2009, under a research permit from the Ethics Committee of Tampere University Hospital. Serial daily morning samples had been collected during hospitalization, and at 2–3 weeks, 6 months, and 12 months after hospitalization. As a negative control, we used urine samples collected from healthy, PUUV-negative volunteers; these volunteers provided informed consent.

### 2.2. Proteins and Antibodies

We set up the LFRET assay using europium-labeled baculovirus-expressed PUUV N (described in [3]) and Alexa Fluor 647 (AF647) labeled protein L (described in [4]). We used IgG-free bovine serum albumin (BSA) from Jackson ImmunoResearch Inc. in the LFRET assays. The primary antibodies employed were mouse monoclonal anti-lambda free light chain (3D12) and mouse monoclonal anti-kappa free light chain antibodies (4C11) from Abcam and HyTest Ltd. and goat anti-human lambda light chain and anti-human kappa light chain antibodies from Southern Biotech. Secondary antibodies were AF680-labeled donkey anti-goat immunoglobulin and IR800Dye donkey anti-mouse immunoglobulin antibodies from LI-COR Biosciences and fluorescein isothiocyanate (FITC)-conjugated goat anti-human IgG (1:100), IgA (1:20), and IgM (1:50) antibodies from Jackson ImmunoResearch Inc.

### 2.3. LFRET Assays

We employed the following protocol: reagents were diluted in Tris-buffered saline (TBS, 50 mM Tris-HCl, 150 mM NaCl, pH 7.4) supplemented with 0.2% BSA (TBS-BSA) and added to a microwell plate. Urine was used undiluted whereas plasma was diluted 1:50 prior to dispensing (10 μL each) onto a 384 microwell plate (ProxiPlate-384 Plus F, Black 384-shallow well microplate from PerkinElmer), after which 10 μL of a solution containing antigen (10 nM) and protein L (50 nM) was added to the reaction. We prepared and analyzed each sample in duplicate, measured the results (TR-FRET values) with a Wallac Victor2 fluorometer (PerkinElmer), and normalized the signals to account for leakage of donor emission to acceptor emission wavelength as described [26].

We compared the LFRET data obtained using urine with those obtained using plasma. Urine and plasma pools from PUUV-seronegative individuals served as negative controls. We analyzed all samples available from each of the 40 patients, including those taken during hospitalization and convalescence (up to 12 months), and a set of PUUV-seronegative healthy donor (*n* = 26) urines.

### 2.4. Immunofluorescence Assay (IFA)

We tested a urine panel (16 samples from four patients, three samples during hospitalization and one after discharge for each patient) using an in-house immunofluorescence assay (IFA) based on PUUV-infected acetone-fixed Vero E6 [17,27]. Briefly, the urine samples were diluted 1/2, 1/5, and 1/10 in phosphate-buffered saline (PBS) and incubated for 1 h at 37 °C. The slides were washed three times with PBS prior to the addition of FITC-conjugated goat anti-human IgG (1:100), IgA (1:20), and IgM (1:50) antibodies diluted in PBS. After 30 min at 37 ° C, the slides were washed three times with PBS, once with Milli-Q water, and air-dried. Finally, a Shandon^TM^ immuno-mount served to attach the cover glasses. PUUV-negative urine served as a negative control, and PUUV-positive serum as a positive control.

### 2.5. Western Blot (WB)

We examined by WB a panel consisting of consecutive urine samples collected during the hospitalization and convalescence of eight patients. The samples (30 μL of urine) were separated on ready-made 4–20% sodium dodecyl sulfate–polyacrylamide electrophoresis (SDS-PAGE) gels (Bio-Rad, Helsinki, Finland) and wet-blotted onto nitrocellulose membranes following standard protocols. The membrane was blocked using 3% skimmed milk in TBS, and sequentially probed with the following antibodies: 1:1000 diluted goat anti-human lambda light chain (Southern Biotech, Birmingham, AL, USA) followed by 1:10,000 diluted IRDye800-labeled donkey anti-goat immunoglobulin (LI-COR Biosciences, Lincoln, NE, USA) and 1:1000 diluted mouse anti-human free kappa light chains (Abcam, Cambridge, MA, USA) followed by 1:10,000 diluted AF680-labeled donkey anti-mouse immunoglobulin (LI-COR Biosciences). All antibody incubations were in T-TBS (TBS+0.05% Triton-X-100) with 3% skimmed milk; all washing steps were with T-TBS. The membranes were washed three times with TBS prior to recording the results with an Odyssey Infrared Imaging System (LI-COR Biosciences).

### 2.6. Immunoprecipitation (IP) of FLCs and PUUV N Protein

We coupled mouse monoclonal anti-kappa (clone 4C11) and anti-lambda (clone 3D12) free light chain antibodies (both from HyTest Ltd., Turku, Finland) to Pierce NHS-activated Magnetic Beads (Thermo Fisher Scientific, Vantaa, Finland) following the manufacturer’s protocol with 400 μg of antibody per 500 μL of activated bead slurry. The coupled beads were used to immunoprecipitate PUUV N protein with free light chains from serum. Briefly, we made four pools of plasma and urine—from the following: 1) healthy volunteer samples (*N* = 2); 2) old immunity, specifically samples collected 6–12 months post PUUV infection (*N* = 10); 3) PUUV patient samples collected at 2–4 weeks after onset of fever (*N* = 11); and 4) PUUV patient samples collected during hospitalization, specifically 4–10 days after onset of fever (*N* = 11). We incubated the pooled samples (10 μL of pooled plasma diluted in 500 μL of TBS with 1 mg/mL of BSA; or 25 μL of pooled urine diluted in 500 μL of TBS with 1 mg/mL of BSA; or 1 mL of urine pH-adjusted by adding 50 μL of 1 M Tris-HCl, pH 8.0) with the antibody-coupled beads for 20 min at room temperature (RT), washed the beads four times with T-TBS, incubated the bead-bound FLCs with 800 μL of AF647-labeled PUUV N protein diluted (approximately 0.5 μg of labeled N protein per reaction) in TBS with 0.5 mg/mL of BSA for 15 min at RT and washed the beads five times with T-TBS. The bound proteins were eluted by a Laemmli sample buffer and analyzed by SDS-PAGE.

### 2.7. Purification and Analysis of Binding Specificities of Free Light Chains in Urine

We coupled 0.5 mg of anti-lambda free light chain (3D12) and 0.5 mg of anti-kappa free light chain antibodies (4C11) (both from HyTest Ltd.) to CNBr-activated Sepharose^TM^ 4B (GE Healthcare, Helsinki, Finland) following the manufacturer’s protocol, with the exception that we used only 0.5 mg of antibody per ml of activated Sepharose instead of the recommended 5–10 mg/mL. The antibody-conjugated Sepharose was packed into Poly-Prep chromatography columns (Bio-Rad). The columns were washed with PBS (10 column volumes) prior to loading the samples. The FLCs were purified from the urine of healthy volunteers (15 mL) and from the urine of patients with acute-phase PUUV infection (2 mL). Prior to purification, the pH of urine samples was adjusted by the addition of 1 M Tris-HCl, pH 8.0 to yield 25 mM, after which the samples were centrifuged for 10 min at 5000 relative centrifugal force (RCF). The samples were loaded into the anti-FLC columns by gravity flow and were initially passed through the anti-kappa FLC column, and then through the anti-lambda FLC column. After loading, the columns were washed with 20 column volumes (CVs) of PBS. The bound FLCs were eluted with 10 CVs of 40 mM citrate-phosphate buffer pH 3. The eluates were collected into a tube containing 1.3 CVs of 1 M Tris-HCl pH 9. The fractions were analyzed by SDS-PAGE and WB as described above, with the concentration estimates based on WB band intensities.

We next performed LFRET using the PUUV-positive kappa light chain fraction, using the FLC fraction from negative urine samples as controls. We used a 1/2 dilution (in TBS-BSA) of the acute-phase FLC fraction, whereas the negative control was undiluted. The negative control was used undiluted since the concentration of the FLCs from negative control urine based on WB analysis was roughly half of that of the PUUV-positive kappa light chain fraction. The antigen and protein L concentrations were as above, and the sample and FLC volumes were 10 μL. Since at this point the FLC fractions were still in the elution buffer (+ added Tris-HCl) not routinely used in the LFRET assay, we also performed another experimental setup with the FLC buffer changed into TBS. In this setup, TBS (without BSA) served as the dilution buffer also for the antigen and protein L (at the concentrations indicated). We also wanted to study if FLCs purified from PUUV-positive patients could induce a positive LFRET signal in the presence of intact antibodies, and we used serum from PUUV-negative volunteers as the source of intact Igs. We tested in triplicate the following samples of three types: (1) 5 μL of PUUV-antibody negative serum, 10 μL acute-phase kappa light chain fraction (in TBS), and 5 μL of antigen + protein L; (2) 5 μL of PUUV-negative serum, 10 μL of TBS, and 5 μL of antigen + protein L (a negative control); and (3) 5 μL of PUUV-positive serum, 10 μL of TBS, and 5 μL of antigen + protein L (a positive control).

### 2.8. Reference Diagnostic Methods

PUUV infection was diagnosed using an IgM enzyme immunoassay [17]. Clinical and laboratory data for the patient samples were acquired earlier using standard methods [28].

### 2.9. Statistics

Correlation between the non-normally distributed LFRET and the albumin levels in individual urine samples during acute PUUV infection were assessed by Spearman’s rank correlation coefficient using SPSS software version 24 (IBM).

## 3. Results

### 3.1. Detection of Hantavirus-Specific Antibody Responses from Urine Using LFRET

We had previously observed extremely high signals in the LFRET assay (as described in [3]) for some patients with acute PUUV infection. Since protein L binds kappa light chains, we hypothesized that a portion of such a signal could be attributed to FLCs) binding the antigen. Since serum FLCs are excreted via urine, we decided to analyze a previously obtained collection of urine samples from 40 patients with acute PUUV infection. The panel consisted of consecutive samples collected during hospitalization in addition to samples taken at convalescence 2–3 weeks after discharge (corresponding to 3–4 weeks after onset of fever) and at 6 and 12 months after resolution of fever. As a negative control, we had a set of urine samples (*n* = 26) from PUUV-IgG-negative healthy donors.

To account for any inter-run variation, we chose not to use a fixed cut-off value. Rather, in each run we analyzed two pooled negative samples and set the cut-off for positive signal at 2.1 times the mean of these negative pools. All of the negative controls (*n* = 26) induced urine-LFRET (uLFRET) values below this threshold. The specificity of the assay was therefore 100%. In the samples taken during hospitalization, 40/40 patients produced a positive uLFRET result (i.e., at least one of the samples for each patient yielded a positive uLFRET result). As summarized in Table 1, on days 4–5 after onset of fever, 96% (23/24) were uLFRET positive; on days 6–7, 100% (30/30); at 1–2 weeks, 100% (*n* = 20/20); and at 2–4 weeks, 85% (*n* = 29/34). Interestingly, at 6 months post infection, 41% (11/27) and at 12 months, still 42% (11/26) of patients were uLFRET positive. The corresponding plasma-LFRET (pLFRET) values are shown in Table 1. The measured uLFRET and pLFRET values are shown in Figure 1.

We have previously demonstrated that the LFRET signal follows the serum antibody concentration in a dose-dependent manner [4]. To show a similar effect for urine, we measured the uLFRET signal from serially diluted pooled negative and positive samples. The results showed the uLFRET (normalized uLFRET values) signal to decrease along with the dilution of the urine sample, as illustrated in Appendix A.

After the removal of statistical outliers (*n* = 5/160), the uLFRET and pLFRET scores of patient samples showed a moderate positive correlation (*r* = 0.49). A detailed timeline showing the corresponding uLFRET and pLFRET scores for individual patients (*n* = 10) is presented in Appendix A, and a plot of the uLFRET and pLFRET scores of the entire sample set is shown in Appendix A. In most cases, the samples collected during hospitalization produced the highest uLFRET and pLFRET scores; the recorded scores declined towards the convalescent phase of the disease.

### 3.2. IgG, IgM, and IgA Antibodies to PUUV in Urine by IFA

Acute PUUV infection is characterized by AKI of varying severity, including proteinuria [10,11,12] that can lead to leakage of Igs in urine. Therefore, we examined with IFA a set of urine samples to assess the extent to which intact Igs contribute to the observed TR-FRET signals. From a subset of four patients, we had a time-series (four samples/patient) from the day of hospitalization to 2–3 weeks after discharge. Most, but not all (14/16), urine samples contained PUUV-IgG. Five samples also had PUUV-IgA, but none had PUUV-IgM. Table 2 summarizes the IFA and uLFRET results. We then studied the serially collected urine samples by WB under non-reducing conditions, using anti-light chain antibodies to detect both free and heavy chain-associated light chains. The result (Figure 2) showed that intact Igs were detectable to a varying extent during hospitalization but less so during convalescence. The blots also showed the concentration of FLCs to be very high during hospitalization and to decline towards convalescence (Figure 2). Furthermore, the WBs for both kappa and lambda chains indicate the FLCs to be more abundant in urine than intact Igs. However, when we compared the induced uLFRET scores to the quantity of intact antibodies and FLCs (patients #1–#4 in Figure 2) in the same samples, we found no obvious correlation between the uLFRET score and the intensity of the bands in WB.

### 3.3. Both Kappa and Lambda FLCs Specifically Bind PUUV N Protein

We performed an immunoprecipitation assay to study if FLCs in the plasma and urine of patients with acute PUUV infection could bind the antigen (PUUV N protein) used in the LFRET assay. FLCs from the pooled urine and plasma of healthy volunteers and PUUV patients were immunoprecipitated using magnetic beads coated with either a kappa or lambda FLC-specific monoclonal antibody. For the initial experiment, we used the pooled urine and plasma of healthy volunteers and acute-phase (collected at 4–10 days after onset of fever) samples of PUUV patients. The beads with FLCs were then incubated with AF647-labeled PUUV N protein, and the bound proteins separated in SDS-PAGE were visualized using an infrared scanner. The results (Figure 3A) showed that patients with acute PUUV infection possess PUUV N protein-binding kappa and lambda FLCs. Urine samples showed only background binding to beads, likely indicating the lower amount of FLCs in urine. For the next experiment, we increased the amount of urine (from 25 μL to 1 mL) as the source of FLCs and included pools of patient samples collected at different time points (4–10 days, 2–4 weeks, and 6–12 months after onset of fever). The results (Figure 3B) showed that both the urine and plasma of PUUV patients collected during hospitalization and around two weeks from hospitalization (corresponding to 2–4 weeks after onset of fever) contained kappa and lambda FLCs that specifically bind PUUV N protein.

Next, we aimed to demonstrate the ability of the acute-phase urine-derived kappa FLCs to bind PUUV N protein in LFRET assay. We purified urinary kappa FLCs from patients with acute PUUV infection, and from healthy controls, and analyzed the eluates in WB (Figure 3C). Based on WB, the positive sample contained approximately twice the amount of FLCs when compared with the negative sample (Figure 3B).

Hence, in the following experiments we used the acute-phase FLCs at a 1:2 dilution, and undiluted control FLCs in two distinct assay setups. In the first setup, we used purified kappa FLCs in place of urine. This approach produced a positive TR-FRET signal (normalized test value 28) with acute PUUV FLCs while the negative-control FLCs produced no signal. In the second approach, we “spiked” purified acute PUUV FLCs in PUUV-negative serum. PUUV-positive serum served as a positive control and PUUV-negative serum as a negative control. The sample containing spiked PUUV FLCs produced a 22% higher signal than the sample containing only the negative control serum, whereas the positive control produced a 43% higher signal than the negative control.

### 3.4. Lack of Correlation Between LFRET Signal and Proteinuria

Proteinuria is a well-known sign in acute PUUV, and we hypothesized that the increased uLFRET signal could potentially associate with the amount of protein in urine. To test this, we assessed the correlation between uLFRET and albumin in urine samples of acute PUUV (days 5 to 16 after the onset of symptoms, *n* = 43) by using Spearman’s rank correlation coefficient. Interestingly, no correlation was observed (*r* = −0.31, *p* = 0.841), indicating that the increase in the uLFRET signal was not associated with the extent of albumin in urine. To investigate the kinetics of the uLFRET signal in relation to albumin output, we plotted these factors together at different times post onset of fever (Figure 4). While both factors were elevated at early time points (days 4 to 6), the uLFRET signal followed an increasing trend thereafter, whereas albumin levels declined due to normalization of kidney function.

## 4. Discussion

We had previously observed extremely high test values in the LFRET assay described in [3] for some patients with acute PUUV infection. Since protein L binds kappa light chains, we hypothesized that some of the signal could be attributed to FLCs bound to the antigen. This prompted us to study whether detection of antigen-specific FLCs could serve as a surrogate for antibody detection. This would benefit from the fact that FLCs, unlike intact antibodies, are constantly excreted via urine [25]. To test this hypothesis, we studied the utility of urine as clinical material in LFRET-based serodiagnostics. To this end, we had an extensive and thoroughly characterized sample panel from hospitalized patients with PUUV infection, including urine and plasma samples. We indeed demonstrated that anti-PUUV N antibodies can be detected in urine and we furthermore demonstrated the presence of antigen-specific FLCs in urine.

While the results did indicate that uLFRET works efficiently, the proteinuria of hantavirus disease urged us to determine whether and to what extent intact antibodies contribute to the observed uLFRET signals. For this purpose, we analyzed a set of urine samples in IFA and WB. We followed up the kinetics of intact Igs in the urine of four patients, which all showed intact Igs in urine during hospitalization, and even at 2–3 weeks after discharge. However, IgA antibodies were found in the urine from two of four patients, indicating PUUV-specific IgG dominance over IgA in urine. Two samples that were IgA, IgG, and IgM negative for PUUV in IFA were nevertheless positive in uLFRET, supporting the notion that FLCs targeting viral antigens contribute to the uLFRET signal. Since the LFRET assay is semi-quantitative [3], we compared the IFA titers and normalized uLFRET results. Some, but not all, of the uLFRET values were consistent with the IFA results, indicating that intact Igs also play a role in uLFRET signal formation.

This lends support from the frequent uLFRET positivity of the 6- and 12-month post-infection samples, when the renal function of the patients has normalized. However, as slight proteinuria can be present even five years after PUUV infection [29], the possibility exists that the LFRET signals are due to intact Igs. Indeed, while analyzing urine samples in WB, we observed both intact Igs and FLCs not only in the acute phase, but also up to 6 or 12 months post infection for some patients. While the WB analysis of past infection samples showed intact Igs at levels barely detectable, the FLCs were abundantly and indisputably present. Because urine albumin levels correlate with kidney leakage, which could cause the leakage of intact Igs into urine, we calculated the correlation between uLFRET values and urine albumin levels. By this we wanted to determine if the uLFRET score would correlate with kidney leakage in the acute stage. The fact that we found kidney leakage not to correlate with the uLFRET score supports the hypothesis that uLFRET signals are mainly caused by FLCs rather than intact Igs. We used the serial dilutions of pooled urine to demonstrate the dose-dependence of the uLFRET signal (Appendix A), which further indicates specific binding. By comparing the pLFRET and uLFRET results, we observed only a moderate positive correlation between the signals (Appendix A), supporting the hypothesis that FLCs contribute to LFRET signals. To further address the role of FLCs in the generation of uLFRET signals, we purified kappa FLCs from the urine of PUUV seropositive versus seronegative individuals and demonstrated that the FLCs purified from PUUV patients induce the LFRET signal even when mixed with serum. The IP experiments with FLC-specific antibodies provide the strongest evidence demonstrating specific antigen binding by FLCs. We could demonstrate that both kappa and lambda FLCs of PUUV patients specifically bind PUUV N protein. The IP results appear to be consistent with the uLFRET scores; the samples collected during hospitalization and approximately two weeks from discharge produced the strongest signals in both.

While the FLCs likely contribute to the LFRET signal also in plasma/serum, the FLC-attributed signal is likely less prominent, since the concentration of intact Igs is up to 1000-fold higher in plasma/serum. In PUUV patient urine, as indicated by our WBs, the FLCs dominate in quantity over intact Igs, due to which the role of FLCs in LFRET signal generation from urine is likely considerable. In the serum/plasma LFRET, IgG depletion permits case definition between acute phase and past infection [3]. However, we cannot use a similar depletion in the case of FLCs and thus uLFRET cannot at this stage differentiate between acute infection and past immunity. We speculate that uLFRET diagnostics would thus be ideally suited for the diagnosis of chronic infections, autoimmune diseases, or allergies, in which merely demonstrating an antibody response could aid in diagnosis.

## 5. Conclusions

We herein demonstrated that the antibody response against PUUV can be measured at a high sensitivity and specificity with the unorthodox POC assay LFRETusing urine as sample matrix. We also demonstrated that acute orthohantavirus infection induces antigen-specific FLCs; to our knowledge this is a novel observation in an infectious disease context. It is likely that the same holds true for other infectious pathogens. We anticipate great potential in the utility of FLCs in the diagnosis of infectious diseases, and potentially also in autoimmune diseases and allergies.

## Figures and Tables

**Figure 1 viruses-11-00809-f001:**
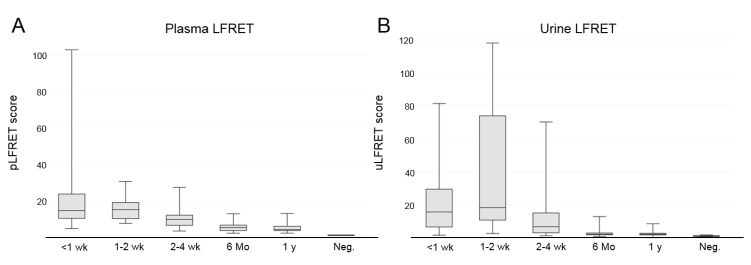
Boxplots of LFRET signals obtained from plasma (pLFRET) and urine (uLFRET): (**A**) pLFRET values of all plasma samples from the 40 PUUV-patients taken at different time points (<1 week, 1–2 weeks, and 2–4 weeks from onset of fever) of the disease, as well as at the convalescent phase (6 months and 1 year post infection); (**B**) uLFRET values of all urine samples from the 40 PUUV-patients taken at different time points (<1 week, 1–2 weeks, and 2–4 weeks from onset of fever) of the disease, as well as at the convalescent phase (6 months and 1 year post infection). The y-axis represents normalized TR-FRET values divided by the average of TR-FRET signals induced by the negative-control samples (LFRET score of >2.1 is considered positive). The whiskers represent minimum and maximum values and the boxes represent the 25–75% quartiles.

**Figure 2 viruses-11-00809-f002:**
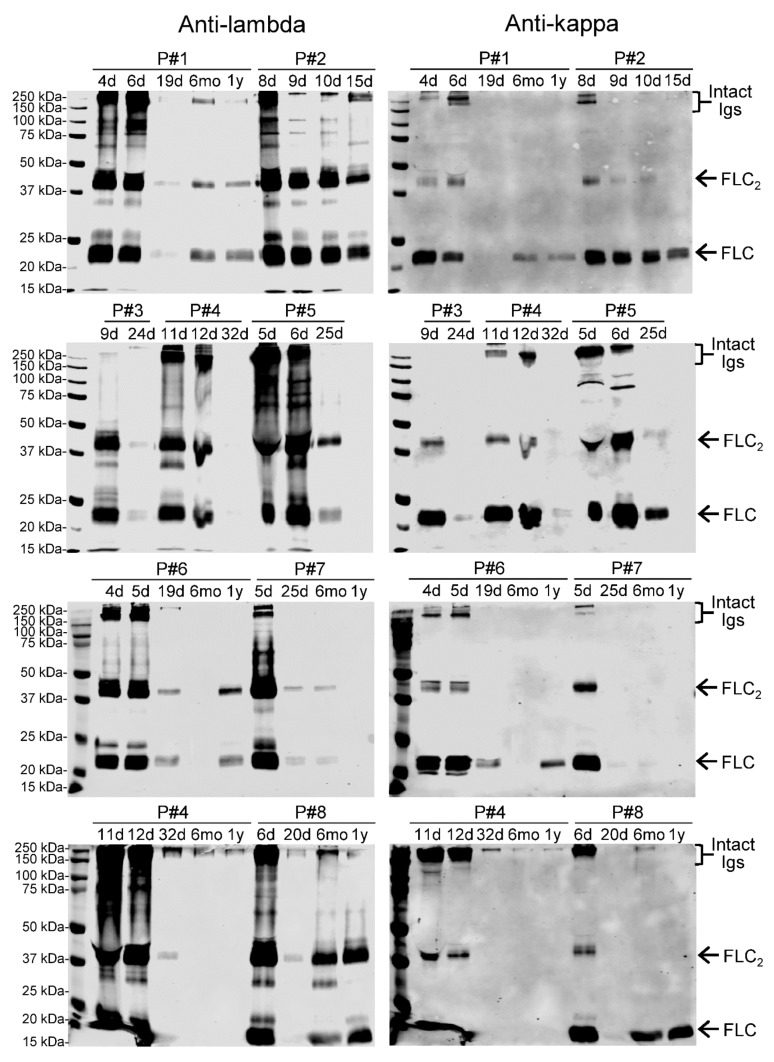
Western blot analysis of urine samples. Urine samples (30 µL/lane) separated in SDS-PAGE under non-reducing conditions were blotted onto nitrocellulose and sequentially immunoblotted with anti-lambda and anti-kappa light chain antibodies. Patients #1–#4 are included in Table 2. The panels on left show the results for anti-lambda light chain staining (probed first, detected using IR800-conjugated secondary antibody) and the panels on right show anti-kappa light chain staining (detected using AF680-conjugated secondary antibody). Molecular weight markers (Bio-Rad, precision plus protein dual color standards) are always the leftmost lane. FLC (free light chain) indicates monomeric and FLC_2_ dimeric FLCs. All detections were performed using an Odyssey Infrared Imaging System (LI-COR Biosciences).

**Figure 3 viruses-11-00809-f003:**
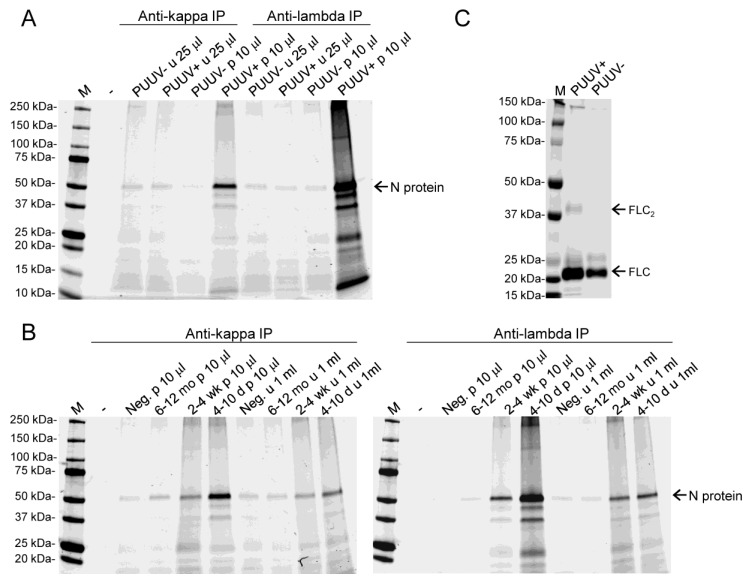
Immunoprecipitation (IP) of PUUV N protein using FLCs and purification of free kappa light chains from urine: (**A**) Monoclonal antibodies against free kappa (clone 4C11) and lambda (3D12) light chains were conjugated to Pierce NHS-activated magnetic beads (Thermo Fisher Scientific) and used for IP of FLCs and PUUV N protein. The left lanes show anti-kappa IP and the right lanes anti-lambda IP results of AF647-labeled PUUV N protein. The samples are indicated above each lane (u stands for urine and p for plasma); the PUUV+ pools were represented by samples collected during hospitalization. The bound PUUV N protein was visualized using an Odyssey Infrared Imaging System (LI-COR Biosciences) at IR700 channel after SDS-PAGE separation; M represents the molecular weight marker (Bio-Rad, precision plus protein dual color standards); (**B**) The experimental setup described in panel **A** was used for IP of AF647-labeled PUUV with FLCs from the urine and plasma of healthy volunteers and PUUV patients (three time points). The left panel shows the results of IP with anti-kappa coated beads and the right panel IP with anti-lambda coated beads. The samples are indicated above each lane (u stands for urine and p for plasma) (**C**) Eluates (20 μL/lane) from monoclonal (clone 4C11) free kappa light chain antibody coupled CNBr-activated Sepharose 4B columns after passing through urine from patients with acute PUUV infection (PUUV+, 2 mL) and healthy volunteers (PUUV−, 15 mL) were analyzed by western blotting using a polyclonal anti-kappa light chain antibody. Detection was performed used an Odyssey Infrared Imaging System (LI-COR Biosciences), M represents the molecular weight marker (Bio-Rad, precision plus protein dual color standards).

**Figure 4 viruses-11-00809-f004:**
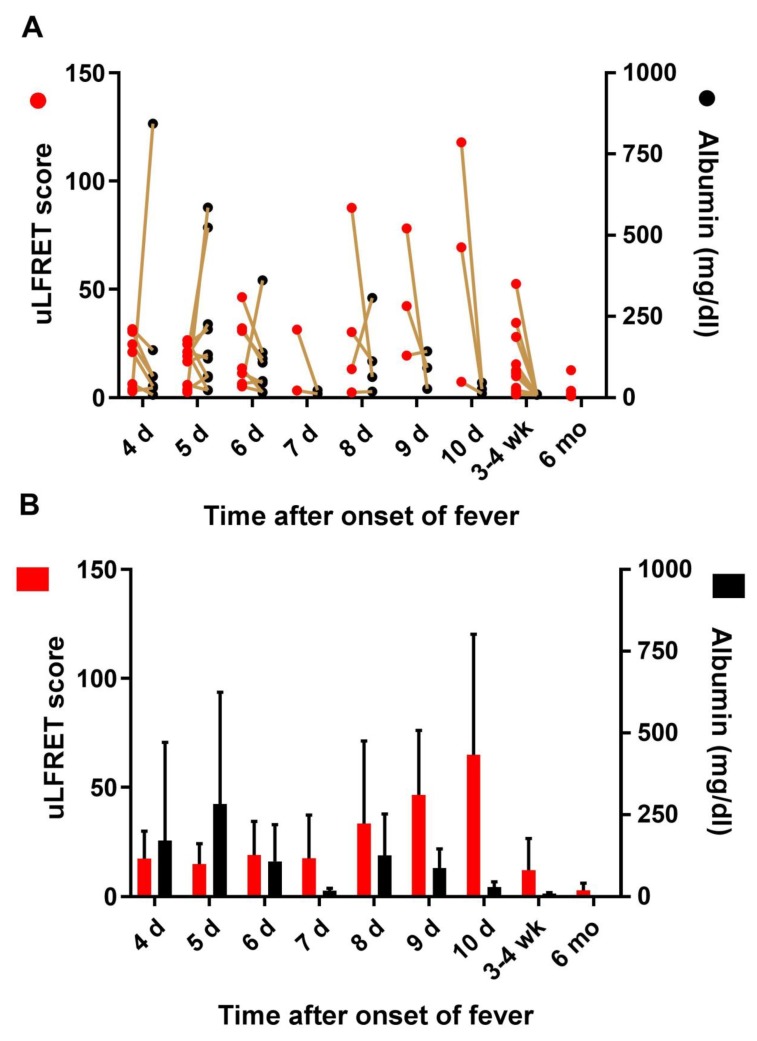
Kinetic analysis on the level of the uLFRET signal in relation to proteinuria. The uLFRET scores were plotted together with albumin levels in urine samples at 4–10 days after onset of fever (acute stage) together with recovery (3–4 weeks) and 6 months (only uLFRET values were available at 6 months). Only time points where N was at least three were included. The y-axis, “uLFRET score”, represents normalized TR-FRET values divided by the average of the TR-FRET signals induced by the negative-control samples (signal/background): (**A**) The individual uLFRET and protein values in each sample are shown and connected by a line; (**B**) Average uLFRET signals and level of proteinuria (albumin) at different time points are shown as mean + standard deviation.

**Table 1 viruses-11-00809-t001:** Sensitivity rates of the uLFRET and pLFRET assays at different time points (4–5 days, 6–7 days, 1–2 weeks, and 2–4 weeks from onset of fever) of the disease and at the convalescent phase (6 and 12 months post infection).

Time after Onset of Fever	4–5 d	6–7 d	1–2 wk	2–4 wk	6 mo	1 y
**Number of urine samples**	24	30	20	34	27	26
**uLFRET positive**	23	30	20	29	11	11
**Sensitivity**	95.8%	100.0%	100.0%	85.3%	40.7%	42.3%
**Number of plasma samples**	23	29	20	34	22	25
**pLFRET positive**	23	29	20	34	21	25
**Sensitivity**	100.0%	100.0%	100.0%	100.0%	95.5%	100.0%

**Table 2 viruses-11-00809-t002:** Presence of PUUV antibodies in urine as determined by immunofluorescence assay (IFA) and uLFRET. The number indicated in parentheses under uLFRET represents the normalized TR-FRET value divided by the average of the TR-FRET signals induced by the negative-control samples (signal/background). The numbers ranging from 2 to >10 in parenthesis under IFA IgG, IgM, or IgA indicate the antibody titer according to IFA. The threshold for positivity in uLFRET is 2.1 times the background signal. * = number of days from the onset of fever until sampling.

Patient 1	IFA IgG	IFA IgM	IFA IgA	uLFRET
5d *	neg	neg	neg	pos (19)
6d	pos (>10)	neg	neg	pos (46)
7d	pos (5)	neg	neg	pos (32)
21d	pos (2)	neg	neg	pos (4)
**Patient 2**				
8d	pos (>10)	neg	neg	pos (88)
9d	pos (>10)	neg	neg	pos (78)
10d	pos (2)	neg	neg	pos (7)
24d	pos (2)	neg	neg	pos (11)
**Patient 3**				
7d	pos (>10)	neg	pos (>10)	pos (13)
8d	pos (5)	neg	pos (2)	pos (42)
9d	pos (5)	neg	neg	pos (69)
24d	pos (2)	neg	neg	pos (52)
**Patient 4**				
10d	pos (>10)	neg	pos (2)	pos (118)
11d	pos (5)	neg	pos (2)	pos (109)
12d	pos (>10)	neg	pos (2)	pos (105)
32d	neg	neg	neg	pos (70)

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
