# Peer review of "Urine and Free Immunoglobulin Light Chains as Analytes for Serodiagnosis of Hantavirus Infection"

_viruses, 2019, doi:10.3390/v11090809_

Round 1
Reviewer 1 Report
Comments
Authors have previously developed a homogeneous wash-free immunoassay based on time-resolved Förster resonance energy transfer (TR-FRET) for serodiagnosis of Puumala orthohantavirus (PUUV) infection. The assay is based on simultaneous binding of an immunoglobulin molecule to its antigen and protein L and referred to as LFRET. This study showed that urine can be used as sample matrix in LFRET-based serodiagnostics and that antigen-specific free light chains (FLCs) is induced by acute PUUV infection and secreted in urine. The findings show great potential of urine and FLCs in diagnosis of infectious diseases.
Major points
1) Authors described that “the antibody response against PUUV can be measured...at a high sensitivity (100% in the acute phase) and specificity (100%)” in the Conclusions section. However, Table 1 showed that 1 of 24 samples at 4-5 days and 5 of 34 samples at 2-4 weeks after onset of fever were uLFRET negative. The results indicate that the sensitivity in the acute phase is not 100%. Please show detailed information about the uLFRET negative samples and provide the appropriate period for diagnosis. Show range of hospitalization period as well.
2) It is important to compare uLFRET scores and the quantity of intact antibodies and FLCs in urine. Therefore, it is preferable to use the same samples in the experiments in Table 2 and Figure 2. At least, show uLFRET scores of samples in Figure 2.
3) Show the ability of the acute-phase urine-derived FLCs to bind PUUV-N protein by co-immunoprecipitation.
4) In lane 2 of Figure 3A, a small amount of PUUV N protein was co-immunoprecipitated with FLCs from patients with acute infection other than PUUV. How much are the pLFRET and uLFRET values in samples from those patients with acute infection other than PUUV? The examination is important for the evaluation of specificity of LFRET.
Minor points
1) line 123: The description “two during hospitalization, one at 2-3 weeks, one at 6 Mo and one at 12 Mo” is correct? Information of samples in Table 2 is different.
2) line 133: The number of patients is different with that in Figure 2.
3) line 146, 150, 233, 259: The “µ” is not shown in some places, although that may be attributed to my computer.
4) line 148: Change “TFRET” to “LFRET”.
5) line 148: 4) PUUV patient sera with “high” LFRET values
6) line 198: Show cut-off value.
7) Figure 1: The figures A) and B) are in reverse order, when compared to legend.
8) line 222: “five” also PUUV-IgA
9) Figure 1, Table 2, Figure 4: The uLFRET values in Figure 1, Table 2, and Figure 4 seem to be calculated in a different way. Calculate the values in the same way.
10) Table 2: Change “FRET” to “uLFRET”. Add explanation about asterisk.
11) Figure 2: Add indicator of intact Igs. Add explanation about FLC2.
12) line 175-180, 273-276: It is difficult to understand the purpose of the experiments and the interpretation of results. Add or change explanations. The “control FLCs” in line 275 is “negative control FLCs”? The “control FLCs” gave 22% higher signal as compared to PUUV negative control serum?
13) line 317-318: Why does “the fact” suggest uLFRET signals to be mainly caused by FLCs rather than intact Igs? Add more explanation.
14) line 327: Reference 30 is the same as reference 3. Change “(30)” to “(3)” in line 327 and remove reference 30 from the reference section.
15) line 327-328: Difficult to understand. Add more explanation.
Reviewer 2 Report
Comments to Editor:
The manuscript by Hepojoki et al. presents a new application for FRET-based technology to detect acute Puumala orthohantavirus infection using urine. Previously, they demonstrated the utility of this technique using serum as the testing matrix (Hepojoki et al. (2015) J. Clin Micro, reference #3). Here, they apply the same technique to urine and investigate whether the urine signal occurs due to the presence of intact antibodies and/or free light chains.
The motivation of this manuscript is to apply a novel diagnostic tool for detection of the Puumala virus N protein to a non-invasive specimen type, such as urine. A major challenge to the development of any diagnostic tool is to determine whether 1) it can specifically detect the microorganism of interest and 2) the limits of detection (ie – specificity and sensitivity).
I agree with the authors – the development of a quick, cheap, and fast test for the diagnosis of acute hantavirus infection is something needed in the field. The added benefit of using a non-invasive specimen matrix is especially enticing, too.
I think that the authors have made a good first start with this manuscript, but some additional clarity and controls are needed in the manuscript. Unfortunately, in its current state the manuscript is not acceptable to publication in Viruses. However, if the authors can provide the additional data, and provide some requested clarification, then I think that it can be re-considered for publication.
Major Concerns:
-Table 1: The authors demonstrate that they can detect an anti-N signal in urine and plasma. What is the limit of detect for this signal? Did the authors try diluting either the urine/plasma or nucleocapsid protein to demonstrate a dose-response curve to the N signal?
-The x-axis scale bar in Figure 1A and 1B is unclear. How were these scores calculated? As supplementary data, it would be informative to see the curves for individual patients over time.
-Figure 1A and 1B: The change in plasma and urine LFRET values is interesting and suggests specificity of the diagnostic test. I was wondering what the correlation was between these values are for individual patients? If plasma and urine LFRET values were plotted on x and y axes, do the authors observe any correlations?
-Table 2: The values in the table need some additional clarification. How were the IFA IgG/igM/IgA tests determined to be positive? What is the significance for the numbers in parentheses? Likewise, what is the scale for the FRET values? How can I determine a strong signal from a weak signal? Also, what is the * in patient #1, 5d mean?
-Figure 2: As currently presented this figure does not appear to contribute much value to the manuscript. The figures demonstrate that lambda and kappa light chains can be detected in patient urine, but no appropriate loading controls are used. Urine concentrations can vary based on food and liquid input. So any changes in light chain intensities in these blots could be due to changes in urine concentrations, and not due to changes in light chain concentrations. Unless I am mistaken, these blots show total changes in light chain intensities, and are not displaying any anti-N specific light chains.
-Figure 3A: This is some nice data. However, again, it’s difficult to interpret the results without a loading control for the plasma. It could be that plasma from the PUUV- patients was more dilute than plasma from the PUUV+ patients, especially since there appears to be weak binding to the N protein for PUUV- patients.
-Figure 3B: The significance for this figure is unclear. In general, the results and Materials and methods sections describing these results are also very unclear. It seems that the authors tested various conditions to purify light chains from urine and that ultimately, they used the purified urine light chains to test for binding to fluorescent N protein. However, Figure 3B appears to only demonstrate that it is possible to purify light chains from the urine of PUUV- and PUUV+ positive patients. Are the light chains in this figure specific to the N protein? Again, no loading controls are used, so it’s hard to evaluate whether light chain concentration is increased in PUUV+ patients. Lines 270-6 are especially unclear, and while the authors appear to have done a lot of work, it’s difficult to evaluate the results.
Figure 4: I appreciate that the authors demonstrate a lack of relationship between the proteinuria and anti-N signal that they detect.
Line 319-20: “Affinity purified antigen-specific FLCs” Unless I am mistaken, I did not see this data in the manuscript. I was hoping that Figure 3B would demonstrate antigen-specific FLCs from the urine. But instead, it demonstrates that FLCs have been purified from urine and does not provide data as to whether these FLCs are N-specific.
Discussion – the authors do not cite any references from the field. What other diagnostic tests exist for hantaviruses? Are other pathogens detects using FLCs from urine? What’s already known/published about FLCs in urine?
Minor Concerns:
-How was Puumala infection diagnosed for this subset of patients?
-Line 295: “Based on our earlier observations…” is unclear.
-Line 301: “antiviral antibodies” is not a clear statement. This diagnostic test is detecting anti-nucleocapsid light chains. It is not detecting the virus.
-Line 327-8: The last sentence is difficult to understand. The grammar appears to be a bit off.
Round 2
Reviewer 1 Report
Comments
Major points
I think the manuscript has been improved. However, authors did not show any responses to “Major points” in the previous comments of reviewer 1. Authors should provide responses to the comments.
Minor points
1) line 316 in the revised manuscript: Volume of PUUV-positive serum is missing.
2) line 389-396: Results of Figure S1 and S2 are not explained in the text.
3) Figure S2: The figure is a duplicate of the figure S3. Replace with correct figure.
4) Figure 3A: It is better to show time point of PUUV+ samples used in figure 3A in figure legend.
Reviewer 2 Report
The authors have carefully reviewed and edited the manuscript. The added analysis, experimentation and clarification have significantly improved it. Some additional minor textual editing might be required before publication, due to the new edits. However, I recommend accepting the manuscript. The editors have addressed my concerns and the additional experimentation improves their conclusions.
